# How Does Carbon Footprint Create Shared Values in the Wine Industry? Empirical Evidence from Prosecco Superiore PDO's Wine District

**Luigino Barisan, Marco Lucchetta, Cristian Bolzonella and Vasco Boatto ***

Department of Land, Environment, Agriculture and Forestry (TESAF), Via dell'Università,
16-35020 Legnaro (Padova), Italy; luigino.barisan@unipd.it (L.B.); marclafleur28@gmail.com (M.L.);
cristian.bolzonella@unipd.it (C.B.)
*** Correspondence: vasco.boatto@unipd.it; Tel.: +39-0438-450475

**Abstract:** Nowadays, the wine sector's entrepreneurs are increasingly aware of the relevance of sustainability representing a crucial point for society, economy and the environment. This paper aims to describe Conegliano Valdobbiadene Prosecco DOCG (Controlled and Guaranteed Denomination of Origin) firms' behaviour focusing on how strategic incorporation of environmentally sustainable practices and social actions contribute to strengthen their competitiveness and deliver shared value for the community. Using partial least squares structural equation modelling (PLS-SEM), survey data from 128 sparkling wine firms are analysed. The results highlight the roles of carbon footprint and employment as drivers in the creation of shared values (SVs), considering the major export markets of Prosecco Superiore DOCG. This empirical evidence may be of interest to firms in the wine sector when considering, in their business decisions, the added commercial value that is derived for the adoption of environmental practices and sustainable social actions. Hence, following this logic, they can manage more inclusive and virtuous paths towards positive social entrepreneurship and an environmental externality to the community.

**Keywords:** Conegliano Valdobbiadene Prosecco DOCG; partial least squares structural equation modelling (PLS-SEM); wine market and community; lower carbon emissions; employment; sustained competitive strategies

## 1. Introduction

One is used to seeing sustainability in a purely ecological and environmental context, but its strategy can also embrace other areas such as the social and economic ones.

With regards to the relationships between the economic value of a firm and its social value, the investments in human capital can generate employment [1,2]. At the same time, considering the environmental subject, the hookup is between the size of a firm and its behaviour: small and medium-sized firms are more sensitive to the environmental problems of microenterprises, while a firm with a large number of employees is more likely to generate positive externalities for the environment [3,4]. In particular, the adoption of practices aimed at reducing the carbon footprint also allows the creation of economic benefits for firms as well as for the environment [5,6]. According to the literature review, several techniques for reducing the carbon footprint in the agriculture fields are mentioned, like diversifying crop rotation and crop rotation systems, use of soil bio-resources, mulching, usage of high analysis fertilizers, contour farming, and no-till farming [7–10].

In the wine sector, environmental sustainability represents one of the critical points for the advancement of knowledge for the growing relevance of innovative and eco-sustainable practices [11].



In particular, the vineyards are not always located in the most suitable areas, often requiring more treatments than expected with pesticides that undermine human health [12] and the environment [13,14], leading to a loss in the economic performance. Moreover, an excessive use of water resources represents a critical issue for viticulture and even for wine production phases [15,16]. Therefore, the calculation of the water footprint represents a useful tool in improving the firm's economic performance and allows it to be more effective as well as to safeguard the water resource for the local community [17,18].

If we consider the social aspect of viticulture, it appears to be the driving force for sustainable social development in particular rural areas [19]. Indeed, viticulture with proper management can produce social development in the same area, but sustainable development requires different factors, low-cost and long-term energy resources without the risk of undesirable social consequences [20]. Finally, according to Pullman et al. [21], the applied social sustainability practices lead to better quality in wine production and in the wine market performance.

In the field of Italian Protected Denominations of Origins (PDO), the Conegliano Valdobbiadene Prosecco DOCG (CVPD) ranked fourth for relevance of DOCG's supply (Controlled and Guaranteed Denomination of Origin) [22]. The CVPD product specifications require grape production to be carried out in a particular area in the Conegliano-Valdobbiadene hills comprising 15 boroughs (8088 hectares) (Figure 1).

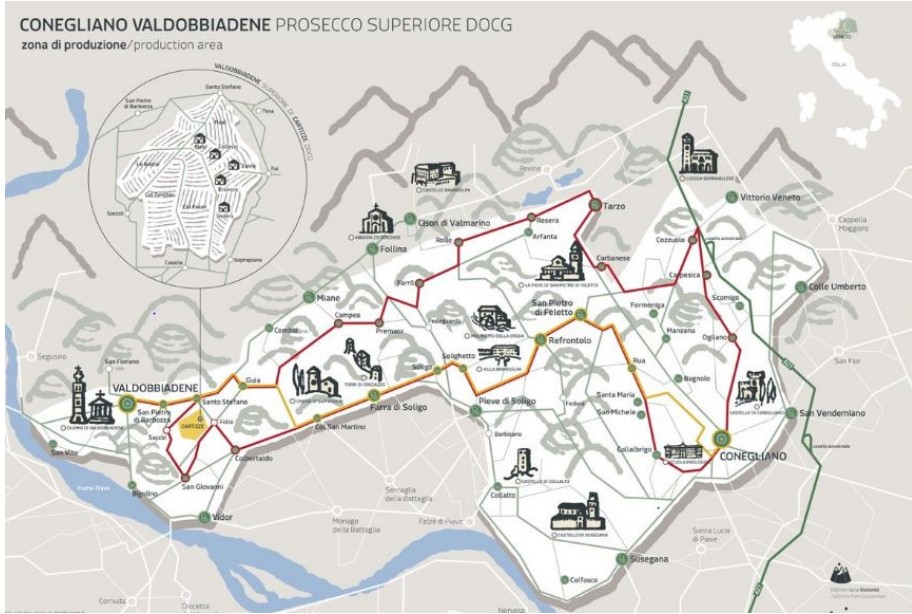

**Figure 1.** Conegliano Valdobbiadene Prosecco DOCG: location of the Protected Denomination of Origin.

Sparkling wine is produced by the Martinotti method while still wine is obtained by a primary fermentation (base wine) which then undergoes a secondary fermentation in pressure tanks. After this, it is bottled with an isobaric filling machine [23].

Currently, the CVPD's production is about 93 million bottles, which represents about 18% of the market share of the total Prosecco (Prosecco DOC, Asolo Prosecco DOCG and CVPD) supply; such a relatively small percentage represents the top quality Prosecco ("Prosecco Superiore DOCG"), which is rooted in a specific tradition, terroir and landscape. The CVPD's production has been supplied to over 130 countries, which is 41% of the exports [24,25].

The CVPD's supply involves 3422 grape-producing firms, 427 plants producing base wine and 185 plants performing second fermentation and bottling. In 2017, the staff employed in the CVPD sparkling wine district showed the presence of 6245 full-time employees, of whom about 52% were in wine-growing firms and 48% were employed in vintners and wine-making houses.

From a sustainable point, the CVPD Consortium has concretely invested in environmental practices and social actions. In particular, five macroindicators (targets) of social and environmental sustainability were identified: protection of health and food safety, preservation of the landscape and biodiversity, creation of a circular economy, reduction of the water footprint and reduction of the carbon footprint [26]. In the latter concept, concrete developments were highlighted, such as greater use of low-emission tractors with a 27% share of tractors in the area, while, taking into consideration the energy production from renewable sources, this reached 6500 MW annually in 2017, generating about 20% of the total energy required for the production process.

This paper aims to describe CVPD firms' behaviour, focusing on how strategic incorporation of environmental practices and social actions contribute to strengthen their competitiveness and deliver shared value (SV) for the community. More specifically, according to Porter and Kramer [27], the paper addresses two related issues by assuming SV as a crucial concept for pursuing firms' sustainability when considering:

- how environmental key drivers are involved in an SV increase on the major Prosecco Superiore DOCG's export markets
- how critical social factors are involved in an SV increase on the major Prosecco Superiore DOCG's export markets

The paper is structured as follows: In the first part of the article, the theoretical framework and the methodology used to explore SV are introduced. In the second part, the major findings from data analysis are presented. Finally, the concluding discussion of the work is reported.

## 2. The Theoretical Framework and Hypothesis

As far as the inner model's specifications are concerned, on the top-left side of Figure 1, the firm structure is represented as an exogenous construct and modelled as a prediction of the level of employment. The employment construct has been considered as a latent endogenous variable and has dual relationships (functioning as an independent construct) causing value creation on the export markets for Prosecco Superiore DOCG and predicting the commitment of the side to reduce the carbon footprint.

Likewise, the firm's competence levels are considered as exogenous variables in predicting the commitment to reducing the carbon footprint. The carbon footprint construct has been considered as an endogenous latent variable and has a relationship, functioning as an independent construct, indicating causal relationship of value creation in the targeted export markets. The increased value generated on export markets is a latent endogenous variable that arises from both the benefits delivered by increased employment and the commitment to reduce the carbon footprint.

This design is in line with both similar research and strategies implemented in the wine industry [26,28–32].

In their review of literature on social sustainability and economic performance management practices, Luthans and Youssef [33] argued how the company that invests in human capital generates a benefit that will come back to it in terms of economic performance. The empirical evidence currently faced by the wine industry suggests that the main areas of social sustainability are: increasing employment, improving competencies, improving the quality of life for employees (training programs and safety courses), supporting the local community (creating and sharing the value of the territory by involving its inhabitants) together with the improvement of economic performance indicators [22,34]. Considering these theoretical approaches and empirical evidence for the wine sector, the following hypothesis is formulated:

**H1.** *Employment has a positive influence on the creation of SV on the major Prosecco Superiore DOCG's export markets.*

As far as the concept of involvement in environmental sustainability performance and the creation of shared economic value in the wine industry are concerned, many studies and empirical evidence showed their relevance from different viewpoints. They concern the efficient use of water and management [35], recycling of materials and their management and treatment [36], energy use and reduction of greenhouse gas emissions [37], sustainable use of crop protection products; land use and environmental preservation, conservation of biodiversity, care and protection of the landscape [38]. The customers seem to have a growing interest in sustainable production, carbon footprint reduction, fair trade, and so forth. The study by Klohr et al. [39] showed that, in Germany, wine consumers are divided into four classes and one of these (wine experts and more habitual consumers) is willing to pay more for a sustainable wine. Consumers tend to choose sustainable wines and even organic wines mainly from sustainable practices adopted [40]. In Austria, there is a willingness to pay up to €1.44 per bottle more for a sustainable one than a conventional one so the organic one receives a higher price and recognition in value [41]. The Canadians expressed preferences in wine consumption without labels that demonstrate specific environmental sustainability [42]. At the same time, however, they expressed a willingness to pay a premium of 65% more for a wine that has a label that defines in detail its certification of sustainability and low environmental impacts. Meise et al. [43] argued that in Switzerland, consumers prefer sustainable products if well explained on the label, even ignoring the lower price; for example the wines labelled as "sustainable" or "eco-friendly" have a 76% preference for consumers compared to those without any specification of sustainability or environmental friendliness, while if there is no information, consumers will prefer products at a lower price. Considering this evidence, we, therefore, hypothesize the following:

**H2.** *The involvement in reducing carbon footprint has a positive influence on the creation of SV on the Prosecco Superiore DOCG's major export markets.*

The two hypotheses reached above are aligned with the theoretical framework developed by Porter and Kramer (2011), who define SV as the set of policies and operational practices that strengthen the competitiveness of firms while improving the economic, environmental and social conditions of the communities in which they operate. The concept of SV defined that the success of a firm is not only due to the result of its economic performance. Indeed, it depends also on a series of environmental conditions that allow it to exist, to produce, to have success and growth. At the centre of the concept of SV is the identification and expansion of the link between environmental and social well-being and economic performance. This can be reached in three different ways: creating new products and new markets, redefining the concept of productivity in the value chain and creating districts to support the firm's competitiveness. Creating SVs from reinventing wines and markets also allows us to focus on internal operations that reduce costs, access to production inputs, quality of supply and the productivity of input factors while, at the same time, improving the firms' perceived image value and reputation and their mutually loyal relationship with the customer. This allows them to explore how an unmet target need leads to incrementing revenue and profits for firms and to delivering both positive social entrepreneurship and environmental externality to the community.

The hypothesis expressed above, according to Porter and Kramer's conceptual approach (2011) and empirical analysis from the annual report on CVPD [26,27], leads to the following path model (Figure 2). The firm structure is represented as an exogenous construct and modelled to predict the level of employment. The employment construct has been considered as a latent endogenous variable and has a dual relationship (functioning as an independent construct), causing value creation on the export markets for Prosecco Superiore DOCG and predicting the commitment for reducing the carbon footprint.

Likewise, a firm's competence levels are considered as an exogenous variable in predicting the commitment to reduce the carbon footprint. The carbon footprint construct has been considered as an endogenous latent variable and has a relationship, functioning as an independent construct,

indicating causal relationship in value creation by means of the targeted export markets. The increased value generated on export markets is a latent endogenous variable that arises from both benefits delivered by increased employment and the commitment to reduce the carbon footprint.

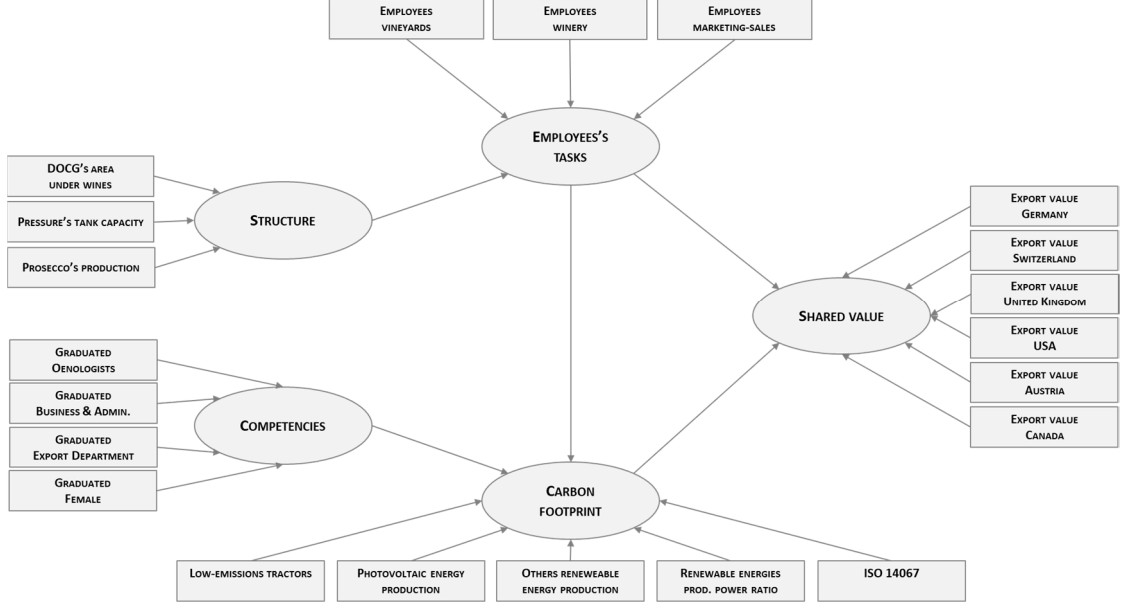

**Figure 2.** Prosecco Superiore DOCG's SV path model: relationships and constructs. Source: Economic Report of Conegliano Valdobbiadene Prosecco DOCG's Wine District, 2018.

## 3. The Conegliano Valdobbiadene Prosecco DOCG Case Study

This study focuses on two key drivers of social and environmental sustainability involved in creating SV: the increase in employment and the activities aimed at reducing the carbon footprint of the firms' wine production chain.

In the wine market, the SV, which incorporates social, environmental and economic value, shows a latent variable that is regarded as being a composite factor and that can be measured indirectly through a set of elements. Hence, the composite factor can be taken into account as a proxy for the concept under assessment [44]. As argued by Henseler et al. (2016), PLS path modelling represents an approach that can be applied to measure artifacts, where different elements are combined to form a new construct [45]. By using these (formative) measures, it is possible to capture relevant specific aspects of the construct and their weight within the construct (which acts as the beta coefficient in the regression) [46–48]. Interestingly, as reported by Speklé and Verbeeten (2014), artifacts are used in management accounting for measuring business performance where its different aspects are tied to form a single construct [49].

Among those listed in the research, some examples could find possible positive repercussions for the CVPD's case study like, among others, the process improvements, a reputation for work excellence, the efficiency of operations and the growth of the personnel's employment and professional levels.

According to Porter and Kramer (2011), the study's approach aims at both reconceiving products and markets and redefining productivity in the value chain [27], focusing on the growth in value of export markets that arises from the development benefits originated by the delivered firm's Prosecco DOCG wines.

## 4. Materials and Methods

### 4.1. Research Approach and Design

The data used in the research come from the database of the C.I.R.V.E. (Research Centre for Viticulture and Enology) [50]. The survey was addressed to the firms' owners or the persons in

charge of the social, environmental and marketing issues in 2017. The survey's sample has been identified with 128 sparkling wine firms, representing about 69% of all listed bottlers members of CVPD's Consortia. The only firms selected in the survey were fully integrated models, where the three production phases performed by the same firm were considered (i.e., grapes production, production of base wine and secondary fermentation and bottling). Then, addressed data collection issues were performed. The suspicious response pattern was examined (i.e., visual inspection of the straight line, anomalies related to extreme pole responses by using descriptive statistics). The presence of outliers was interpreted in the context of the study.

In order to investigate the hypothesized model, concerning the roles of Prosecco Superiore DOCG in creating SV in the wine sector, a structural equation analysis was considered a suitable one to test models holistically [51,52].

Partial least squares analysis (PLS) with the software SmartPLS 3 was used to analyse the dataset [53]. A Structural Equation Modelling (SEM) approach allowed the testing of multiple dependent and independent constructs in a single research model. With regards to the type of analysis selected, the study has considered the partial least squares (PLS), a variance-based SEM technique. According to Wold's approach (1975) (originally called NIPALS, nonlinear iterative partial least squares), PLS is characterized in the maximization of the variance of the dependent variables explained by independent variables [54].

PLS-SEM technique was preferred over the covariance-based model (CB-SEM) for the following reasons [55,56]. First, the paper aimed to identify key driver constructs of SV. Second, the exploratory characteristic of the research dealt with possible extensions of the theory developed by Porter, using specific constructs suitable for the wine sector. Third, the model specifications consisted of formative constructs. Fourth, although the survey sample included about 70% of all the CVPD's wine producers, its size was relatively low. Fifth, the items in data collected were non-normally distributed (i.e., skewness and/or kurtosis).

The study has taken into consideration the applicative approach to the research situation, the regressions using sum scores that equalize the negative weights of the indicators that make up the construct.

### 4.2. Operationalisation of the Constructs

As far as the outer model's specification is concerned, when considering the question of specifying constructs in a reflective and/or formative manner, the latter was chosen [46,47,55]. The choice derives from the managerially oriented business of research aimed at identifying the critical drivers (i.e., proxies) [48,49] that ultimately contribute to creating SVs and obtaining nuances of interest and recommendations for producers of the Prosecco Superiore DOCG sparkling wine district.

The firms' SV was operationalized as a multidimensional component model that comprises five dimensions: "structure", "competencies", "employment", "carbon footprint" and "shared value".

The firms' structure was defined by three items: Protected Denominations of Origins' surface under Glera vines, storage capacity in pressure tanks for the production of sparkling wines, amount of total bottle sales in the CVPD's portfolio of sparkling (i.e., Prosecco Superiore, Superiore di Cartizze and Rive) and semisparkling (i.e., frizzante) wines [57]. Among these, the former amounts to around 94% of the CVPD's supply.

The firms' competencies construct was measured by graduated personnel employed across crucial phases of the vine and wine supply chain (i.e., oenologists, business and administration and export department) and by female graduated employees [22].

A measurement of the firms' employment was used while taking into consideration the number of full-time employees per task (i.e., winegrower, vintner and oenologist, administrative and export personnel, logistic personnel, commercial director) when these were carried out in the three main phases of the wine production chain (vineyard, cellar, sales and logistics) [22,58]. Indicators of employment were based on metric data.

Carbon footprint was operationalized by four items: the use of low-emission tractors in the vineyard (number); the generation of electricity from renewable energy sources in the cellar (photovoltaic); the generation of electricity from other renewable energy sources (i.e., biomass, hydroelectric, solar thermal, etc.); and the adoption of the carbon footprint certification of products (ISO/TS 14067) [59]. For this construct, metrics data were used.

SVs performance was defined as a result of firms' employment, carbon footprint and Prosecco Superiore DOCG's export markets by value (Germany, United Kingdom, Switzerland, United States, Austria, Canada). Metric data were considered as indicators of market export competitiveness.

Finally, data analysis was carried out in the following steps: First, the outer model was analysed by evaluating the formative measurements of the variables. Second, the inner model was analysed by evaluating the path coefficients and the explanatory power of the hypothesized model.

## 5. Results

### 5.1. Description of the Sample Firms

In Table 1, the sample characteristics in terms of structural and sales variables are reported. Considering the CVPD's surface under vines of firms' sample, small firms prevail (49.2% of the total), followed by medium firms (38.3%), while the relatively big ones make up the rest (12.5%). The firms' economic results by bottles sold showed a non-normal distribution of sales value, showing in the [25] majority of cases relatively small–medium size firms (85%), below 1 million bottles, with a smaller number of firms reaching 1 million bottles and above (15%). Firms' core competencies, analysed by phases of the production cycle, exhibited an evenly balanced number of employees across major phases of the supply chain (from grape production to wine production, to marketing and sales). The firms of the sample pursued an export strategy focused mainly to the top six export markets, where Germany, Switzerland, United Kingdom and the United States followed by Austria and Canada are playing key roles (77% of the total export value) [25].

**Table 1.** Sample data descriptions.

| Variables | Percentage | Mean | St.dev | Min | Max |
|---|---|---|---|---|---|
| **Vineyard surface** *(hectares):* | | | | | |
| **Small firms** *(below 7)* | 49.20% | | | | |
| **Medium** *(from 7 to 25)* | 38.30% | | | | |
| **Big** *(over 25)* | 12.50% | | | | |
| **Pressure tanks capacity** *(hectoliters):* | | | | | |
| **Small firms** *(below 200)* | 51.5% | | | | |
| **Medium** *(from 200 to 2000)* | 35.00% | | | | |
| **Big** *(over 2000)* | 12.50% | | | | |
| **Size** *(bottles sold):* | | | | | |
| **Small firms** *(150,000)* | 68.30% | | | | |
| **Medium** *(from 150,000 to 500,000)* | 17.10% | | | | |
| **Big** *(from 500,000 to 1 million)* | 7.30% | | | | |
| **Very big** *(over 1 million)* | 7.30% | | | | |
| **Vineyard employees** *(number)* | | 2.7 | 2.795 | 0 | 20 |
| **Cellar employees** *(number)* | | 3.1 | 6.314 | 0 | 53 |
| **Adm. and marketing employees** *(number)* | | 2.6 | 5.196 | 0 | 44 |
| **Prosecco Superiore DOCG sales** *(thousands of bottles):* | | 393 | 1479 | 15 | 15,400 |
| **Market share on export markets** *(%)* | | 24.90% | 26.70% | 0.00% | 95.00% |

Source: Own elaboration on C.I.R.V.E. database.

### 5.2. The Construct Measurements

To assess the significance of the path's coefficients, an accelerated bias-corrected bootstrap with 5000 samples was applied. A two-tailed testing on a significance level of 0,1 was chosen. In Table 2,

the formative measurements evaluations of the CVPD shared value model are reported. It summarizes the results of the formative constructs showing latent variables scores and p-values.

**Table 2.** Evaluation of the formative measurements.

|  | Weights | P-Values |
|---|---|---|
| DOCG's area under vines -> Structure | 0.429 | 0.000 |
| Pressure tanks' capacity -> Structure | 0.429 | 0.000 |
| Prosecco's production -> Structure | 0.429 | 0.000 |
| Employees - Marketing & Sales -> Employees - Tasks | 0.396 | 0.000 |
| Employees - Winery -> Employees - Tasks | 0.396 | 0.000 |
| Employees - Vineyard -> Employees - Tasks | 0.396 | 0.000 |
| Export value - Austria -> SE-Shared Value | 0.229 | 0.000 |
| Export value - Canada -> SE-Shared Value | 0.229 | 0.000 |
| Export value - Germany -> SE-Shared Value | 0.229 | 0.000 |
| Export value - Switzerland -> SE-Shared Value | 0.229 | 0.000 |
| Export value - United Kingdom -> SE-Shared Value | 0.229 | 0.000 |
| Export value - United States -> SE-Shared Value | 0.229 | 0.000 |
| Graduated - Business & Admin. -> Competencies | 0.286 | 0.000 |
| Graduated - Female -> Competencies | 0.286 | 0.000 |
| Graduated - Oenologists -> Competencies | 0.286 | 0.000 |
| Graduated - Export department -> Competencies | 0.286 | 0.000 |
| ISO/TS 14067 standard -> Carbon Footprint | 0.308 | 0.000 |
| Low-emission tractors -> Carbon Footprint - | 0.308 | 0.000 |
| Other renewable energy -> Carbon Footprint | 0.308 | 0.000 |
| Photovoltaic energy prod. -> Carbon Footprint | 0.308 | 0.000 |
| Renewable energy–Green power ratio -> Carbon Footprint | 0.308 | 0.000 |

As far as significant levels are concerned, it was found that all of the indicators were significant whereas all the dimensions analysed showed positive significant weights. In particular, the construct measurements results showed higher latent variable scores for the structure (0.429) and the employment tasks (0.396), followed by carbon footprint, competencies and shared value.

### 5.3. The Hypothesis Tested

From the examination of the inner model results reported in Table 3, it is shown that all the indicators' VIF (variance inflation factor) are below the critical value of 5, so no issues of collinearity between the combinations of endogenous constructs were highlighted.

**Table 3.** Prosecco Superiore DOCG's modelling the shared value: bootstrapping results.

| Paths | Path Coefficients | P-Values | VIF | $f^2$-Sizes |
|---|---|---|---|---|
| Carbon Footprint -> SE-Shared Value | 0.314 | 0.069 | 1.775 | 0.148 |
| Competencies -> Carbon Footprint | 0.405 | 0.084 | 2.672 | 0.122 |
| Employees - Tasks -> Carbon Footprint | 0.340 | 0.038 | 2.672 | 0.086 |
| Employees - Tasks -> SE-Shared Value | 0.547 | 0.000 | 1.775 | 0.448 |
| Structure -> Employees - Tasks | 0.736 | 0.000 | 1.000 | 1.179 |

Considering the analysis of the model's in-sample predictive power ($R^2$ square), it indicates levels of accuracy between moderate and substantial of the endogenous latent variables, ranging from 0.498 of carbon footprint, to 0.541 employees' tasks and to 0.624 shared value.

When considering $f^2$ values for all combinations of endogenous constructs and corresponding exogenous ones, some aspects of interest emerged. On the one hand, both competencies and employees' tasks have roughly a medium-sized effect on carbon footprint, equal to 0.086 and 0.122, respectively. On the other hand, carbon footprint has a medium effect on shared value (0.148), where tasks have large effects (0.448). As expected, the firms' structure has a large effect on tasks (1.179), more interestingly,

from the analysis that shows the four formative driver constructs that ultimately influence shared value, wherever tasks have the strongest total effect on it (0.654), followed by structure (0.481) and carbon footprint (0.314). Besides, competencies have a larger total effect on carbon footprint (0.405) than employees' tasks (0.340). In particular, all the relations reported in the inner model were significant, albeit with different levels of magnitude.

As far as the analysis of the results regarding the hypotheses of the study was concerned, because both the path coefficients between employees' tasks and carbon footprint on Prosecco Superiore DOCG's export market performance are significant, hypotheses $H_1$ and $H_2$ can be accepted. In particular, it was shown that the SVs created on the main export markets of Prosecco Superiore DOCG are maximized by the increase of investments in human capital, measured by employment levels, whereas this report showed a significant effect between moderate and robust (0.563; p-value = 0.000).

Likewise, there is a significant effect between the firms' implementation of investments and practices aimed at reducing carbon footprint and the creation of SVs on wine export markets; however, this report showed a lower magnitude and a more moderate impact on the SVs of wine exports (0.314; p-value = 0.069).

These empirical results, through the positive effects of employment, the reduction of the ecological footprint and the value generated on foreign markets by the Prosecco Superiore DOCG, show the central role of these dimensions in the SV performance creation process.

Table 3 additionally shows that employees' tasks have moderate effects on carbon footprint (0.340). $f^2$ values of 1.179 indicate that structure has a relatively large influence on employees' tasks while competencies have moderate effects on carbon footprint ($f^2 = 0.086$).

## 6. Discussion and Conclusions

This empirical study has delved into the relevance of specific environmental and social sustainability drivers in the creation of SVs on major export markets of Prosecco Superiore DOCG. These findings and their implications for theory, firms and the Protected Denominations of Origin are discussed.

From the standpoint of the results, concerning the conceptual framework adopted in the study, it appears to be in agreement with Porter and Kramer's [27,60] theory derived from the firms' specific strategies aimed at creating SVs by reconceiving wine production and markets and redefining productivity in the value chain, through a logic that integrates the improvement of ecological and social performances into business management.

In this context, the study has tested two hypotheses: the impact of firms' employment on the creation of social value on the Prosecco Superiore DOCG's major export markets (Germany, United Kingdom, Switzerland, United States, Austria and Canada) and the effect of firms' efforts to reduce the carbon footprint while creating social value and export markets.

The study finds significant support in accepting the first hypothesis and moderate support for accepting the second. As far as the first hypothesis is concerned, the results evidenced how increasing employment has a positive direct influence in creating SVs on the top six Prosecco Superiore DOCG's export markets, leading to the achievement of the most important goals of social sustainability with advantages both for local wineries and for the local community [61]. Moreover, the study explained how, in the firms' sample, carbon footprint reduction showed a positive direct influence in creating SVs on the top six Prosecco Superiore DOCG's export markets by improving the firms' market performances while contributing to lower carbon emissions [59]. Consequently, the results related to the two hypotheses tested in the CVPD illustrate the relevance of roles of employment and carbon footprint in explaining the process that generates shared value performance by putting into practice Porter and Kramer's theory.

Based on our data, some practical tips for both CVPD's and firms' managers can be drawn. Since employment and carbon footprint play major roles in creating SVs, it would be appropriate to increase the knowledge on social and environmental crucial factors that improve economic values at

firms and territorial levels. Otherwise, Prosecco Superiore PDO's Wine District would lose significant opportunities both for the growth of firms and for their impacts on the levels of development on the local community and in the profiles of their environmental sustainability. The results suggest that the CVPD's firms should carefully consider effective policy for further reducing carbon footprint and investing in management personnel, as these have a positive influence on the sales performance in the major export markets.

To the best of our knowledge, this paper is the first, at least in the literature concerning the Italian wine sector, that highlights the role of SVs in the relationship of employment and the carbon footprint with value creation on foreign markets. Although this study contributes to shedding light on the literature on the role of SV creation in the wine sector, it has some limitations that should be taken into account in the interpretation of the results. First, the cross-sectional data used in the research design can limit the validity of the causal relationships found among the model constructs; however, in order to shed more light on these findings, it may be useful to conduct further research considering the development of market, environmental and social sustainability factors over time (i.e., using longitudinal data). Second, the inference of the results concerns the structural, organizational and market context of the Conegliano Valdobbiadene Prosecco DOCG's Wine District; therefore, replication of the investigation in other international wine-growing areas is needed to provide external confirmation of the validity of the results founded. Third, the SV model implemented for the wine sector can be further improved in order to understand better the key components on which the operationalization of the constructs is based. Fourth, considering the literature review carried out for the wine sector, the efforts to understand the links between the creation of social and environmental and market SVs appear at the beginning. Therefore, further research should address a more comprehensive overview on this relevant topic.

To conclude, in the global wine market, it might be argued that valuing the firms' sustainable strategies through activities aimed at improving carbon footprint performance and employment can be actionable strategies for sustaining an adequate competitiveness both for firms and Protected Denominations of Origin.

**Author Contributions:** Conceiving research and designing research framework, V.B. and L.B.; analysing and processing data, M.L., C.B. and L.B.; all authors wrote and reviewed the paper.

**Funding:** This research received no external funding.

**Conflicts of Interest:** The authors declare no conflicts of interest.

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
