# Peer review of "How Does Carbon Footprint Create Shared Values in the Wine Industry? Empirical Evidence from Prosecco Superiore PDO’s Wine District"

_sustainability, doi:10.3390/su11113037_

Round 1
Reviewer 1 Report
This study examines the efforts by wine producers in the Prosecco Superiore PDO Wine District to adopt sustainable practices and add to the local social value of their operations. It explores the question of whether this strategy has beneficial effects for the industry by making the products more attractive in export markets.
the study distinguishes among different types of firms by size of employment and wine making capacity. It then tests two hypotheses about the impact of employment on the creation of social value and export markets, and about the effect of efforts to reduce the carbon footprint of these on employment and export market.
The study finds significant support for the first hypothesis and moderate support for the second.
It will be of interest to firms in the wine industry who consider the commercial value of adding sustainable practices to their business decisions.
Author Response
Thank you very much for the time invested in reading the paper and for providing us with comments and suggestions. They were very useful and considered to improve the paper further.
Reviewer 2 Report
The article is well written and articulate in most respects. The language selected can be difficult to follow (meaning the article is not a casual read) but appropriate to the subject. The subject may have a smaller audience than could be expected for articles focused on carbon footprint analysis (due to the specificity of the topic). That said the article was clear and well founded and provides a reasonable degree of uniqueness to the field.
I would recomend an additional review for grammar and spelling issues
Author Response
1. Thank you very much for the time invested in reading the paper and for providing us with comments, suggestions and scientific points for improving the manuscript. In addition, the paragraphs on the material and method as well as the results have been extended and better structured in order to further improve the readability of the contents.
2. Thank you for noticing. We have done changes now in order to review grammar and spelling issues, and we have sent it again to an academic English editor.
Reviewer 3 Report
The paper deals with the very important and up-to-date topic. Competitiveness on the global wine market is very important from scientific and from a practical point of view.
The main shortcoming of the paper lack of clear performance. The most of the paper is devoted to the literature review. Nevertheless, the authors seem to deal with too many terms, and readers might be confused. Line 43-59, the text is overload with less important facts. Please establish a more clear relation between the literature review and the goal.
Clear use of terms and concepts is necessary. Especially if we have in mind that the paper is focused on two, employment and carbon footprint.
To analyze is the procedure or the method. It can't aim as such. Aim and objectives could be: to quantify, to determinate something (relationship), to describe...
Aim described in the paper (line 92) isn’t the same as the aim in the abstract (lines 12-14)?
The chapter Results is unusually short. It should prevail?
Table 2 has not been described.
Concept of SV has its theoretical background and authors applied an adequate method. But, one would expect authors to use available data to establish the relationship between investments in social and environmental factors and economic performances of the wineries at first. Is this possible?
Discussion and conclusions part should discuss the results alongside set hypothesis, and not to involve new facts or references. Also, this chapter should clearly discuss where the hypothesis accepted or rejected. One would expect authors to provide more practical bits of advice for the businesses.
Technical details:
In the Abstract (lines 18-21), please be more precise.
Figure 1 is hard to read.
Author Response
1. We thank the reviewer for the time invested in carefully reading the manuscript and the suggestions he has given for improving the paper. They were very useful.
2. Thanks for the note. The paper has been revised to make the content argument more effective. The introduction has been reduced (ex. deletion of lines 81-83, etc). Furthermore, it has been simplified by reducing the terms and concepts used. In accordance with the reviewer, the lines 43-59 have been eliminated, making the reading more fluent and linear. The description in the introduction has been improved to better clarify the relationships between the literature review and the aims (see lines 30-54),
3. In order to improve the clarity of the exposition and the definition of the concepts, paragraph 4.2 has been revised in the definitions (i.e. employment and carbon footprint) and better structured in the order of presentation (see lines 232-262).
4. Thank you, it has been corrected (see lines 11).
5. Thank you for letting us know, we have corrected it (see lines 11-14 and 83-85).
6. Results has been deeply revised and better structured, with additional comments on the main results (see lines 264-324). The comments have been improved, including paragraph 5.1, creating and commenting on paragraph 5.2 and expanding the comments on the results of paragraph 5.3.
7. Thank you for letting us know, table 2 is now described (see lines 280-289).
8. We thank the reviewer for commenting on this important point, that we tried to answer in the conclusions (see lines 350-358).
9. Following reviewer suggestion, Discussion and Conclusions has been rewritten alongside the set of hypothesis (see lines 325-379). In the current version, comments on new facts and not suitable references were deleted, while hypothesis tested were clearly presented when accepted (please see lines 339-340). In addition, practical advice to management staff were provided (see lines 350-353).
10. Following the reviewer's comment, lines 18-21 have been rewritten.
11. Figure 1 has now been replaced (see lines 60-61).